# Properties of Potential Plant-Growth-Promoting Bacteria and Their Effect on Wheat Growth Promotion (*Triticum aestivum*) and Soil Characteristics

Elena Voronina [1,2,*], Ekaterina Sokolova [1,2], Irina Tromenschleger [1,2], Olga Mishukova [1,2], Inna Hlistun [1,2], Matvei Miroshnik [2,3], Oleg Savenkov [4], Maria Buyanova [3,4], Ilya Ivanov [4], Maria Galyamova [2] and Natalya Smirnova [4]

[1]  Institute of Chemical Biology and Fundamental Medicine, Siberian Branch of the Russian Academy of Sciences, 630090 Novosibirsk, Russia; sokolovaea2608@gmail.com (E.S.)
[2]  Novosibirsk State University, 630090 Novosibirsk, Russia; listihhe@gmail.com (M.M.)
[3]  Novosibirsk State Agrarian University, 630039 Novosibirsk, Russia
[4]  Institute of Soil Science and Agrochemistry, Siberian Branch of Russian Academy of Sciences, 630090 Novosibirsk, Russia; nat-smirnova@yandex.ru (N.S.)
*  Correspondence: voronina_l@mail.ru; Tel.: +7-91-3930-0326

**Abstract:** Plant-growth-promoting bacteria are an important economic and environmental resource as biofertilizers that can stimulate plant growth and improve agricultural yields. In this study, potential plant growth-promoting bacteria were isolated from soil samples collected in Russia. Strains that manifested active growth on a nitrogen-free medium, the Pikovskaya medium (with insoluble phosphates) and CAS (Chrome Azurol S) agar, were selected for the study. All bacterial isolates were identified by 16S rRNA gene sequencing analysis. Seventeen bacterial isolates of different species were purified and quantified for their ability to grow on nitrogen-free media; dissolve phosphate; and produce ammonium, indole-3-acetic acid, siderophores, and antifungicidal agents. Principal component analysis identified three groups of strains: one with the maximum signs of providing "plant nutrition"; one with signs of "antimicrobial activity"; and a group "without outstanding signs". All 17 strains were involved in experiments involving growing inoculated wheat seeds (*Triticum aestivum*) in pots under natural environmental conditions, and were assessed by their effect on the wheat growth and yield as well as on the chemical composition of the soil. For the "plant nutrition" group, regression analysis revealed a connection between indicators of plant growth, ear length, and ammonium accumulation in the soil. However, in other groups, there were also strains showing a positive effect on plant growth, which suggests the necessity of involving additional factors to predict the ability of strains to affect plants when screened in the laboratory.

**Keywords:** plant-growth-promoting bacteria; *Triticum aestivum*; nitrogen fixation; IAA (indole-3-acetic acid); phosphate solubilization; siderophore production; antifungicidal agents

## 1. Introduction

The global trend of reducing the application of agrochemicals and the transition to organic farming stimulates the search and introduction of new, additional sources of mineral nutrition and alternative methods to increase the productivity of agricultural crops while protecting them from phytopathogens. Currently, the optimal choice is considered to be the use of plant growth-promoting bacteria (PGPB) and PGPB-containing biological products. At the moment, a large number of bacteria which are useful for plants have been described, which belong to different genera such as Bacillus, Pseudomonas, Burkholderia, Enterobacter, Sphingomonas, Klebsiella, Serratia, Rhizobium, and Bradyrhizobium [1].

To collect PGPB strains, in this study, the "citizen science" approach was used. Schoolchildren, along with their mentor teachers, collected soil samples and conducted a

primary selection of strains capable of growing in a nitrogen-free environment. Citizen science programs provide the opportunity for schoolchildren, students, and interested laypersons to actively participate in scientific research [2]. Such programs are important not only from an educational perspective, but also because they enable scientists to broaden the geographic and environmental scale of their observations. Subsequently, soil samples from different regions of Russia and morphologically unique strains were sent to the laboratory, where they underwent further analysis. Since the "citizen scientists" provided a fairly large flow of strains, there was a question of how to select bacteria with the most positive effects on plant growth.

The direct effect of PGPB on plants is associated with their properties, such as an increase in the availability of mineral nutrition elements, including nitrogen, phosphorus, and potassium [3]; the production of metabolites with hormonal, signaling, and other functions that regulate growth [4]; and the induction of mechanisms with systemic resistance to stresses of either an abiotic or biotic nature [5,6]. PGPB can affect plants indirectly as well, by displacing and suppressing the development of phytopathogenic organisms and reducing the content of harmful chemical compounds and heavy metals in the soil [7,8].

There are methods for evaluating the PGPB traits of strains. However, the analysis of many features often raises the question of which properties should be prioritized when selecting optimal strains. To accomplish this, it is important to assess which characteristics can be used primarily to select promising agricultural strains. This study aimed to examine which indicators are the most important for selecting strains that can be used as biofertilizers. Several strains with varying degrees of manifestation of characteristics beneficial to plants were selected, and their impact on the growth of wheat in pots under natural conditions was examined. In addition, the authors intended to assess their effect not only on plant growth and production, but also on the nutrient content of the soil.

## 2. Materials and Methods

### 2.1. Soil Sample Collection

Soil samples were assembled for the civil scientific project "Atlas of Soil Microorganisms of Russia" from four different areas (Moscow, Yakutia, Sakhalin, Novosibirsk). For the collection, the "envelope" method was used, according to which five samples (3–5 m away from each other) from each sampling site were taken and averaged by means of mixing. Samples (5 g) were placed individually in plastic bags and sent to the Institute of Biology Chemistry and Fundamental Medicine (IChBFM, Novosibirsk, Russia) by mail.

### 2.2. Isolation of Bacteria

Soil microbes were isolated from each sample by the spread plate method. Then, 0.1 g of each soil sample was dispersed in 50 mL of sterilized NaCl (0.9%) and thoroughly shaken using a vortex mixer. Afterward, 100 mkl was spread onto corresponding agar plates.

Some of the samples were obtained from participants in the "Atlas of Soil Microorganisms of Russia" project in the form of mixtures of bacteria growing in columns from the Ashby agar. Such bacterial mixtures were purified to pure cultures and transferred to 12–24-well plates for further testing of their properties.

NFB isolation was carried out on Ashby agar plates with 2% glucose and a trace amount of bromothymol blue (BTB) at $30 \pm 1$ °C. After 3 to 10 days of incubation, colonies with changes in the color of the medium were recorded.

PSB isolation was carried out on Pikovskaya's agar medium (PVK) containing insoluble tricalcium phosphate and a trace amount of bromophenol blue at $30 \pm 1$ °C. After 3 to 10 days of incubation, colonies with changes in the color of the medium were recorded.

SPB isolation was carried out on CAS-agar plates at $30 \pm 1$ °C for 48 h. The siderophore producing bacterial colonies showed an orange color around the colony. The CAS assay is based on a siderophore's ability to bind to ferric iron with high affinity. An indication of siderophore production was the color changing from blue to purple (as described in

the traditional CAS assay for siderophores of the catechol type) or from blue to orange (as reported for microorganisms that produce hydroxamates) in halo shapes around the colonies [9].

Different colonies were chosen and purified using the subculture method on the respective agar media to obtain pure colonies. Colony morphology and color were recorded after 24 h of growth. Bacterial identification was initially performed using gram staining reactions, then examined using a microscope.

### 2.3. Molecular Characterization

Next, the molecular identification of the isolates was determined on the basis of 16S rDNA sequence analysis. Bacterial isolates were cultured for 48 h, and the DNA of the isolates was extracted according to the procedure described by Sambrook et al. [10]. The DNA template for PCR amplification was prepared by picking the individual colony of each strain and amplification of 16S rRNA gene. Amplification of the gene was carried out by PCR using 27F (5-AGAGTTTGATCTTGGCTCAG-3) and 1492R (5-GGT TAC CTT GTT ACG ACT T-3). The reaction mixture (25 μL), prepared for full-length 16S rRNA gene amplification, was initially denatured at 94 °C for 2 min. This was followed by 30 cycles consisting of denaturation at 94 °C for 1 min; primer annealing at 52 °C for 1.5 min; primer extension at 72 °C for 2 min; and, finally, extension at 72 °C for 10 min in a thermal cycler. The purification of PCR products and sequencing were carried out at the SB RAS Genomics Core Facility (http://www.niboch.nsc.ru/doku.php/sequest, accessed on 1 September 2023). The 16S rDNA gene sequences of the bacterial isolates obtained were matched with available gene sequences using BLAST (http://www.ncbi.nlm.nih.gov, accessed on 4 September 2023) and aligned by employing the Clustal W program. Phylogenetic trees were constructed using the neighbor-joining method, and molecular evolutionary analyses were conducted using the MEGA X software (version 10.2.5) [11]. The nucleotide sequences of the bacterial strains were submitted to NCBI GenBank.

### 2.4. Quantitative Assessment of Potential Properties That Promote Plant Growth

Bacterial cultures were grown in 50 mL falcon tubes filled with 10 mL LB broth and kept in a shaker at 200 rpm for 48 h. They were diluted to adjust to $10^8$ cfu/mL bacterial solutions with sterile, distilled water.

#### 2.4.1. Solubilization of Insoluble Phosphate

P-solubilization was quantified via the phospho-molybdate blue color method using a spectrophotometer ($\lambda = 882$, Varioskan Flash; Thermo Fisher Scientific, Waltham, MA, USA), as described by Murphy and Riley [12]. For quantitative evaluation, a comparison with the standard curve obtained using a standard solution of potassium phosphate was used. The experiment was repeated twice, with three replicates each, and the mean was calculated.

#### 2.4.2. Production of Ammonia

Bacterial isolates were tested for the production of ammonia in peptone water. Freshly grown cultures were inoculated in 10 mL peptone water in each tube separately, then incubated on a rotary shaker for 96 h at $28 \pm 2$ °C. After incubation, Nesseler's reagent (0.5 mL) was added to each tube. The development of a brown to yellow color indicated ammonia production. The absorbance was measured at 450 nm using a spectrophotometer (Varioskan Flash; Thermo Fisher Scientific). For quantitative evaluation, a comparison with the standard curve obtained using a standard solution of ammonium sulfate was used. The experiment was repeated twice, with three replicates each, and the mean was calculated.

#### 2.4.3. Nitrogen Fixation

The nitrogen-fixing abilities of the bacterial isolates were determined based on their growth on N-free Jensen's medium by repeated culturing [13]. The cooled medium was

inoculated with specific strains, incubated for 2 days, and checked for growth by measuring the optical density (OD) at 600 nm.

### 2.4.4. Production of Indole-3-Acetic Acid

Bacterial isolates were inoculated in sterilized nutrient broth supplemented with 1% tryptophan (precursor for IAA production), then incubated in a shaker for 3 days at 28–30 °C. After the incubation period, the cultures were centrifuged at 10,000 rpm for 10 min before 1 mL of each supernatant was mixed with 2 mL Salkowski reagent (1 mL of 0.5 M $FeCl_3$ in 50 mL of 35% $HClO_4$) [14]. The mixtures were left at room temperature for 30 min. The development of a pink color indicated the production of IAA, and the quantification of IAA was read at 530 nm on a spectrophotometer (Varioskan Flash; Thermo Fisher Scientific). A standard curve was plotted for the quantification of IAA solution and uninoculated medium, with a reagent serving as a control. The experiment was repeated twice, with three replicates each, and the mean was calculated.

### 2.4.5. Siderophore Production

The CAS shuttle assay was carried out to measure the quantitative production of siderophores, in which CAS reagent was added to the culture supernatant at a ratio of 1:1, followed by measurement of absorbance at 630 nm using multimode reader (Varioskan Flash; Thermo Fisher Scientific) [15]. MS media and a CAS reagent, at a ratio of 1:1, were used as references. The percentage of siderophore units produced was calculated using the following formula: % siderophore unit = $[(Ar - As)/Ar] \times 100$, where Ar = absorbance of the reference at 630 nm and As = absorbance of the sample at 630 nm [16].

### 2.4.6. Antifungal Activity against *Fusarium oxisporum* in Dual Culture Plate Method

To assess whether the isolates exhibited any antifungal activity, their effects on the growth of different fungal cultures were measured using the dual-culture technique [17]. All of the selected isolates were screened for antifungal activities against *Fusarium oxysporum* f.sp. cubense (Foc) using potato dextrose agar (PDA) medium. The isolates were spot-inoculated on PDA medium opposite to pathogenic fungi inoculated at the other side of the medium, 3 cm away in distance. The zone between the isolates and fungi indicated antagonistic interaction between them. The antagonistic activity was investigated for 4 to 7 days after incubation at 28 °C in the incubator.

### 2.5. Plant Inoculation and Experimental Design of Pot Trial

The effect of bacterial inoculation on plant growth was studied on "Novosibirsk 31" wheat varieties in a pot experiment under natural environmental conditions. Pots with diameters of 21 cm that could hold 5 kg of soil were used in this experiment. All the selected seeds were surface-sterilized with 1% NaOCl for 90 s and two consecutive rinses in sterile distilled water, followed by air-drying under laminar air flow. Bacterial cultures were grown in 50 mL falcon tubes filled with 10 mL LB broth and were kept in a shaker at 200 rpm for 48 h, then diluted to adjust $10^8$ cfu/mL bacterial solutions with sterile, distilled water. Seeds were coated with culture by immersion in a suspension of bacteria for 120 min. This experiment was carried out with three replications, and the results were compared with control seeds treated with water instead of a bacterial isolate.

The soils were sterilized at a temperature of 121 °C (103 kPa) for 30 min, hermetically incubated at a temperature of 22 °C for 24 h, and autoclaving was repeated. Five seeds were placed in each pot at a depth of 2–3 cm. The pots were exposed to the street in natural environmental conditions. During the growing period, the maximum temperature was 26 °C, the minimum was 3.7 °C, and the average was 17.1 °C. The relative humidity fluctuated in the range of 33–100%; the average was 72.5%. The experiment lasted 90 days, until the grain matured. The experiment was set up as a randomized design, with three biological replications. The agronomic parameters, such as plant height (cm), spike length

(cm), spike weight (g), dry weight (g), 1000-grain weight (g), root length, number of stems, and number of spikelets were measured at the maturity stage.

### 2.6. Soil Analysis

The soils were analyzed for the following criteria with different protocols [18]:

(1) Soil organic carbon (SOC)—0.1–0.2 g soil, reaction with 0.4 N $K_2Cr_2O_7$ in mixture with $H_2SO_4$;

(2) Soil total nitrogen (STN), determined by the Kjeldahl technique—4 g soil, digestion with 20 mL 95.6% $H_2SO_4$;

(3) Available phosphorus (AP)—20 g soil extracted by 0.03 N $K_2SO_4$, 5 min reaction time;

(4) Exchangeable potassium (Ex-K)—5 g soil < 1.0 mm, extracted by 50 mL of $CH_3COONH_4$, pH 7, 1 h reaction time.

The labile forms of macronutrients (N-$NO_3$, N-$NH_4$) were determined by conservative methods described by Maynard and coauthors [19]. Briefly, the quantity of nitrate was determined potentiometrically after the extraction of 2 g of the dry cadaver material via 20 mL of 0.03 M $K_2SO_4$. The ammonium content was determined colorimetrically after extraction of 2.5 g of the cadaver material by 50 mL of the 2N KCl. Each treatment was replicated three times.

Soil humus (SH) content was determined by the Tyurin method, based on the oxidation of soil organic matter with a mixture of potassium dichromate and concentrated sulfuric acid. The soil organic content (SOC) was determined by stepwise loss through the ignition method using 2–4 g soil aliquots [20].

### 2.7. Statistical Analysis

Comparing the groups for statistical differences in the data, the significance was tested using Student's (*t*) test.

A principal component analysis (PCA) was performed to analyze the relationships between soil isolates and parameters measured by tests. PCA was performed using the R procomp() function with standard parameters (https://www.rdocumentation.org/packages/stats/versions/3.6.2/topics/prcomp, accessed on 12 May 2023). The isolates from the soil were grouped into clusters on a two-dimensional graph of two main components: PCA1 and PCA2.

The principal component regression technique was used further for estimating the statistical significance between major components and plant growth parameters. In the analysis, we included the principal components with higher variances (PCA1 and PCA2). The analysis was carried with the lm() function in R (https://www.rdocumentation.org/packages/stats/versions/3.6.2/topics/lm, accessed on 12 May 2023).

## 3. Results

### 3.1. Isolation and Characterization of PGPB Isolates

A total of 378 bacterial strains were isolated from 9 soil samples, which were collected from different sites in Moscow, Yakutia, Sakhalin, and Novosibirsk (Russia). All the bacterial isolates were deposited at the Center of Applied Microbiology of the Institute of Chemical Biology and Fundamental Medicine (IHBFM), Novosibirsk, Russia, and accession numbers for all bacterial isolates were received (accession nos. GMG-01 to GMG-378). Primary screening was performed on Pikovskaya's agar medium, Ashby agar medium, and CAS agar medium (Figure 1) in 12- or 24-well plates. As a result, 173 bacterial isolates were selected, among which 24 solubilized phosphate, 112 isolates were able to grow on a nitrogen-free medium, and 37 strains produced siderophores.

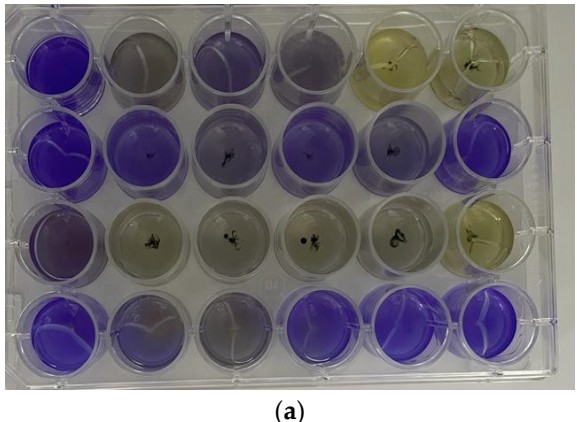
(**a**)

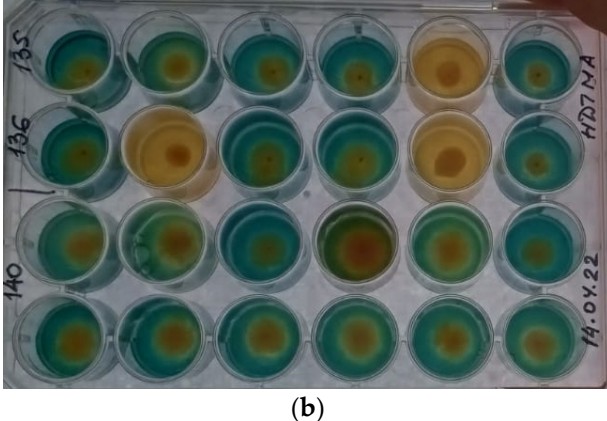
(**b**)

**Figure 1.** Primary screening of phosphate immobilizer strains (**a**) and siderophore-producing strains (**b**).

### 3.2. Identification of Bacterial Isolates

A total of 173 isolates were further subjected to 16S rDNA sequence analysis. Sequence analysis demonstrated that NFB isolates were distributed within five different classes: Gammaproteobacteria (Pseudomonadota)—89, Alphaproteobacteria (Pseudomonadota)—8, Betaproteobacteria (Pseudomonadota)—4, Bacilli (Bacillota)—8, and Actinomycetes (Actinomycetota)—3. Similarly, Gammaproteobacteria (Pseudomonadota) was the dominant phylum in PSB isolates (13), followed by Bacilli (Bacillota)—9 and Betaproteobacteria (Pseudomonadota)—2. In our study, only bacteria belonging to Gammaproteobacteria (Pseudomonadota) were found among the SPB isolates. Isolates representing unique species or unique properties were selected for further research (Supplementary Table S1).

### 3.3. Plant-Growth-Promoting Traits of Bacterial Strains

Seventeen selected strains were evaluated for various PGP traits, such as phosphate solubilization, nitrogen fixation, ammonia, siderophore and IAA production, and antifungal activity test.

Among the 17 bacterial isolates, 5 demonstrated active growth on a nitrogen-free medium, while 6 solubilized phosphate (more than 100 μg/mL). Five isolates were able to produce the phytohormone IAA (more than 5 μg/mL). Eight isolates showed the capacity to produce ammonia (more than 8 μmol/mL), and four isolates were found to be positive for production of iron-chelating siderophores (more than 35% of siderophore units). Antifungicidal activity was also observed for five isolates (Table 1).

**Table 1.** Plant growth promoting traits for 17 isolates.

| No | Strain | Nitrogen Fixation, % | Phosphate Solubilization, μg/ml | Ammonia Production, μmol/mL | Siderophore Production, % | IAA Production, μg/mL | Antifungal Activity, mm |
|---|---|---|---|---|---|---|---|
| GMG_9 | *Rothia endophytica* | 41.8 ± 4.6 | 248.3 ± 35.5 | 8.5 ± 0.3 | 7.1 ± 4.0 | 0.0 ± 0.3 | 0.0 |
| GMG_11 | *Pseudomonas koreensis* | 41.5 ± 10.1 | 294.8 ± 28.1 | 8.0 ± 0.7 | 0.0 ± 0.7 | 0.0 ± 0.1 | 0.0 |
| GMG_14 | *Pseudomonas silesiensis* | 3.9 ± 1.6 | 231.4 ± 39.9 | 4.8 ± 1.3 | 76.8 ± 5.2 | 0.0 ± 0.4 | 16.2 ± 5.1 |
| GMG_20 | *Pantoea agglomerans* | 3.1 ± 1.2 | 105.1 ± 18.8 | 6.5 ± 1.6 | 0.0 ± 0.3 | 6.7 ± 0.3 | 0.0 |
| GMG_21 | *Rhodococcus erythropolis* | 5.5 ± 1.3 | 54.3 ± 7.0 | 4.7 ± 1.4 | 0.0 ± 0.3 | 0.0 ± 0.1 | 0.0 |
| GMG_24 | *Enterobacter cloacae* | 0.6 ± 0.4 | 94.2 ± 18.1 | 8.5 ± 0.2 | 8.4 ± 4.6 | 5.2 ± 0.6 | 0.0 |
| GMG_27 | *Variovorax paradoxus* | 1.9 ± 1.2 | 194.1 ± 24.8 | 6.7 ± 0.8 | 0.0 ± 0.2 | 5.6 ± 0.4 | 0.0 |
| GMG_31 | *Hylemonella gracilis* | 0.4 ± 0.4 | 0.0 ± 10.9 | 8.5 ± 0.5 | 42.8 ± 12.5 | 4.9 ± 0.1 | 0.0 |
| GMG_33.4 | *Agrobacterium arsenijevicii* | 4.1 ± 1.5 | 0.0 ± 8.6 | 5.9 ± 0.6 | 0.0 ± 0.2 | 0.0 ± 0.3 | 11.7 ± 3.2 |

**Table 1.** *Cont.*

| No | Strain | Nitrogen Fixation, % | Phosphate Solubilization, μg/ml | Ammonia Production, μmol/mL | Siderophore Production, % | IAA Production, μg/mL | Antifungal Activity, mm |
|---|---|---|---|---|---|---|---|
| GMG_39.2 | *Azotobacter chroococcum* | 0.9 ± 0.2 | 128.8 ± 20.4 | 6.7 ± 0.8 | 0.0 ± 0.3 | 0.0 ± 0.3 | 0.0 |
| GMG_219 | *Pseudomonas kitaguniensis* | 1.8 ± 0.9 | 75.5 ± 12.9 | 8.0 ± 0.2 | 61.7 ± 5.3 | 0.0 ± 0.2 | 0.0 |
| GMG_234 | *Acinetobacter oryzae* | 9.2 ± 4.1 | 65.8 ± 25.2 | 7.8 ± 0.6 | 3.5 ± 0.7 | 0.2 ± 0.4 | 12.3 ± 2.7 |
| GMG_271 | *Pseudomonas kitaguniensis* | 26.9 ± 3.2 | 151.2 ± 36.3 | 6.4 ± 0.5 | 7.8 ± 3.7 | 12.3 ± 4.6 | 0.0 |
| GMG_278 | *Enterobacter ludwigii* | 31.5 ± 8.6 | 287.3 ± 31.9 | 8.1 ± 0.3 | 7.8 ± 1.2 | 37.8 ± 0.9 | 0.0 |
| GMG_287 | *Rahnella aquatilis* | 9.3 ± 4.0 | 18.3 ± 10.8 | 8.5 ± 0.9 | 0.0 ± 0.4 | 1.6 ± 0.5 | 18.2 ± 3.5 |
| GMG_288 | *Enterobacter amnigenus* | 2.5 ± 1.6 | 52.3 ± 14.3 | 7.5 ± 1.2 | 39.8 ± 2.3 | 3.0 ± 0.3 | 17.5 ± 2.6 |
| GMG_294 | *Rahnella aceris* | 21.2 ± 4.9 | 214.0 ± 33.2 | 7.4 ± 0.5 | 3.1 ± 0.3 | 1.0 ± 0.7 | 0.0 |
| GMG_0 | | 0.0 | 0.0 | 0.0 | 0.0 | 0.0 | 0.0 |

The color shows an increase in the given indicator (red is the highest values, green is the lowest values).

### 3.4. Principal Component Analysis (PCA)

Principal component analysis (PCA) was carried out to cluster strains according to identical indicators of the examined quantitative characteristics. In total, five main components described 97% of the variability of the initial indicators, with the largest contribution from the main component 1 (PCA1), which included the variables "growth on a nitrogen-free medium", "phosphate solubilization", "ammonium production", and "IAA production" and accounted for 39.32% of the total variability.

The second main component (PCA2) grouped the indicators "antifungal activity" and "siderophore production" and accounted for 24.05% of the total variability. In total, the two components explained 63.4% of the variability. All of the variables contributed almost equivalently to the formation of the main components (Table 2), and functionally, as one can see, PCA1 can be characterized as a factor providing "plant nutrition", and PCA2 as a factor exerting "antimicrobial activity".

**Table 2.** Correlation coefficients in the first two principal components.

| | PCA1 | PCA2 |
|---|---|---|
| NFM | −0.50 | 0.27 |
| PM | −0.58 | −0.06 |
| Amm | −0.45 | −0.29 |
| IAA | −0.42 | −0.003 |
| Syd | −0.07 | −0.66 |
| AntiF | 0.12 | −0.63 |

The isolates could be divided into several homogeneous groups depending on the properties they exhibited (Figure 2). Cluster I (9, 11, 271, 278, and 294) contained isolates that manifested the best properties for providing nitrogen and phosphorus availability, as well as auxin production. These strains should have a better effect on plant growth. Microorganisms that actively produced siderophores and antifungal substances were located in cluster IV (14 and 288). These isolates should be more effective at protecting plants from stresses and pathogens. Isolates from cluster II manifested negligible effects in all tests.

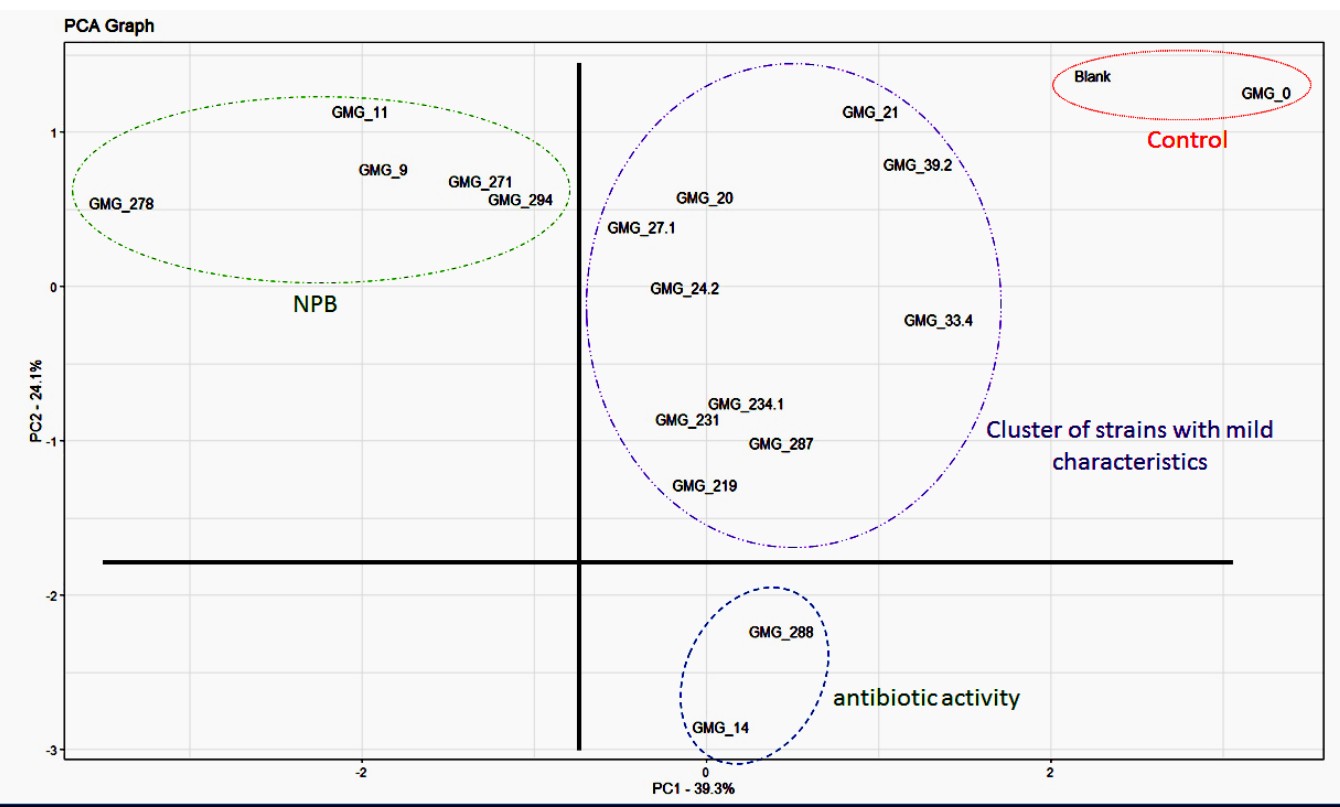

**Figure 2.** Principal component analysis (PCA) among the screened bacterial strains and the PGP traits.

*3.5. Plant Growth Parameters*

All 17 strains were inoculated as "biofertilizers" when planting wheat seeds in the pots. The growing cycle was carried out in outdoor conditions; watering was performed in trays twice a week; and no additional plant protection measures were used.

Plant height, ear length, and weight, as well as chaffless grain weight (without scales), were measured after harvest (Figure 3). All experiments involving bacterial inoculation showed increases in plant growth from 11.3% to 26.5% and ear length from 6% to 38% compared to the control. The weight of the wheat ears showed large variations in values in the experimental groups, from $-6\%$ to 45.5%, but no significant differences compared to the control were found. The weight of dry grain without chaff in the experimental groups ranged from $-12.8\%$ to 22.3% (there are no statistical data, since this indicator was determined in a total sample of three replicates).

Regression analysis showed that PCA1 significantly affected plant growth ($p = 0.00353$) and ear length ($p = 0.000343$). It can be noted that strains from cluster II could also manifest significant increases in the measured parameters. However, in general, the variation in values in this cluster was much higher than in cluster I. Thus, one can assume that the selection of strains based on the indicators included in PCA1 will lead to more predictable positive results related to the influence on plant growth and productivity. However, there are still factors that should move strains with good indicators for impactful plant morphometric signs from cluster 2 to cluster 1. The investigation of these factors could be the subject of further research.

A regression analysis of growth and yield indicators did not find significant correlations for PCA2.

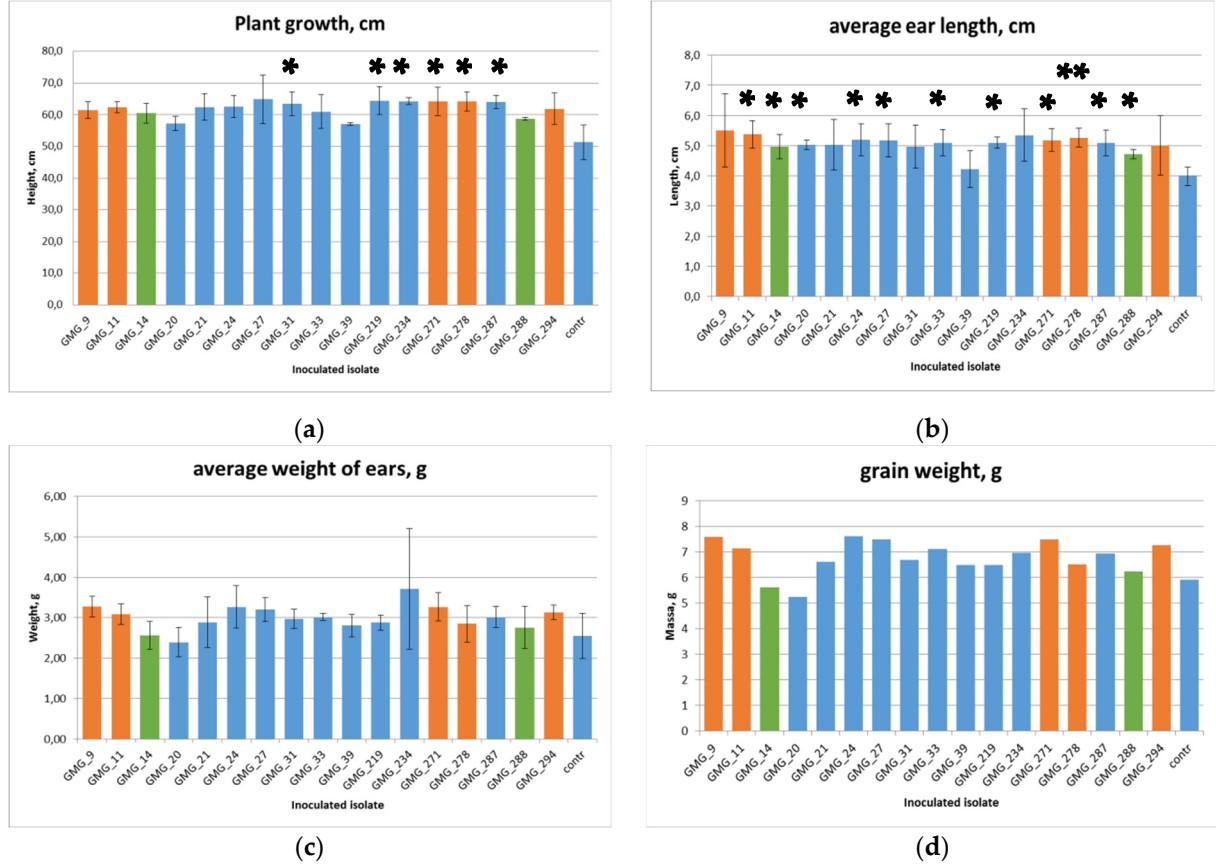

(**a**)                                       (**b**)

(**c**)                                       (**d**)

**Figure 3.** Effect of biofertilizers on plant height (**a**), ear length (**b**), weight (**c**), and chaffless grain weight (**d**). Note: *, differences are statistically significant when $p < 0.05$; **, differences are at the trend level when $p < 0.01$. Orange highlighting—strains from Cluster I, green highlighting—strains from cluster IV.

### 3.6. Soil Characteristics

In each pot, the contents of mobile forms of nutrients of the soil ($NO_3^-$, $NH_4^+$, $P_2O_5$, $K_2O$), as well as the contents of humus and organic carbon, were assessed. The results are presented in Table 3.

**Table 3.** Determination of chemical parameters of soils after growing wheat with the addition of bacteria.

| | Ammonia Nitrogen (N-NH$_4$), mg per kg of soil | Nitrate Nitrogen (N-NO$_3$), mg per kg of soil | Soil Total Nitrogen (STN), % | Soil Humus (SH), % | Soil Organic Carbon (SOC), % | Available Phosphorus (AP), mg per kg of soil | Exchangeable Potassium (Ex-K), mg per kg of soil |
|---|---|---|---|---|---|---|---|
| GMG_9 | 1.81 ± 0.27 | 6.33 ± 1.9 | 0.19 ± 0.02 | 3.58 ± 0.54 | 3.94 ± 0.24 | 0.24 ± 0.01 | 158 ± 15.8 |
| GMG_11 | 1.79 ± 0.27 | 4.29 ± 1.29 | 0.21 ± 0.02 | 3.55 ± 0.53 | 4.11 ± 0.25 | 0.18 ± 0.01 | 158 ± 15.8 |
| GMG_14 | 1.29 ± 0.19 | 3.89 ± 1.17 | 0.2 ± 0.02 | 3.37 ± 0.51 | 4 ± 0.25 | 0.1 ± 0.02 | 158 ± 15.8 |
| GMG_20 | 1.02 ± 0.15 | 3.7 ± 1.11 | 0.2 ± 0.02 | 4.32 ± 0.65 | 3.87 ± 0.25 | 0.18 ± 0.01 | 172 ± 17.2 |
| GMG_21 | 1.4 ± 0.21 | 3.53 ± 1.06 | 0.2 ± 0.02 | 4.34 ± 0.65 | 3.97 ± 0.28 | 0.21 ± 0.02 | 186 ± 18.6 |
| GMG_24 | 0.89 ± 0.13 | 3.36 ± 1.01 | 0.2 ± 0.02 | 4.25 ± 0.64 | 4.01 ± 0.25 | 0.42 ± 0.02 | 158 ± 15.8 |
| GMG_27 | 3.62 ± 0.54 | 3.36 ± 1.01 | 0.18 ± 0.02 | 3.96 ± 0.59 | 3.93 ± 0.27 | 0.11 ± 0.01 | 144 ± 14.4 |
| GMG_31 | 1.55 ± 0.23 | 3.89 ± 1.17 | 0.2 ± 0.02 | 4.08 ± 0.61 | 4.07 ± 0.06 | 0.09 ± 0.02 | 158 ± 15.8 |
| GMG_33.4 | 2.42 ± 0.36 | 3.36 ± 1.01 | 0.2 ± 0.02 | 3.64 ± 0.55 | 4.15 ± 0.26 | 0.13 ± 0.01 | 158 ± 15.8 |
| GMG_39.2 | 1.59 ± 0.24 | 3.7 ± 1.11 | 0.2 ± 0.02 | 3.61 ± 0.54 | 3.97 ± 0.27 | 0.13 ± 0.01 | 144 ± 14.4 |
| GMG_219 | 0.94 ± 0.14 | 3.36 ± 1.01 | 0.2 ± 0.02 | 4.15 ± 0.62 | 4.12 ± 0.25 | 0.12 ± 0.02 | 172 ± 17.2 |
| GMG_234 | 0.83 ± 0.12 | 3.53 ± 1.06 | 0.22 ± 0.02 | 3.83 ± 0.57 | 4.27 ± 0.25 | 0.18 ± 0.02 | 186 ± 18.6 |
| GMG_271 | 0.72 ± 0.11 | 3.2 ± 0.96 | 0.2 ± 0.02 | 3.05 ± 0.46 | 4.01 ± 0.24 | 0.17 ± 0.01 | 158 ± 15.8 |
| GMG_278 | 1.16 ± 0.17 | 4.08 ± 1.22 | 0.2 ± 0.02 | 3.27 ± 0.49 | 3.93 ± 0.26 | 0.14 ± 0 | 172 ± 17.2 |
| GMG_287 | 2.88 ± 0.43 | 3.36 ± 1.01 | 0.21 ± 0.02 | 4.18 ± 0.63 | 4.18 ± 0.26 | 0.1 ± 0.01 | 158 ± 15.8 |
| GMG_288 | 1.31 ± 0.2 | 3.89 ± 1.17 | 0.19 ± 0.02 | 3.5 ± 0.53 | 2.21 ± 0.25 | 0.17 ± 0.01 | 158 ± 15.8 |
| GMG_294 | 1.92 ± 0.29 | 3.53 ± 1.06 | 0.19 ± 0.02 | 4.6 ± 0.69 | 4.02 ± 0.25 | 0.26 ± 0.02 | 158 ± 15.8 |
| GMG_0 | 4.82 ± 0.72 | 3.37 ± 1.01 | 0.22 ± 0.02 | 3.8 ± 0.57 | 4.16 ± 0.24 | 0.2 ± 0 | 200 ± 20 |

Regression analysis of the PCA1 and PCA2 effects on the soil chemical composition revealed that PCA1 significantly affected the amount of soil ammonia nitrogen ($p = 0.00649$). No other significant dependencies were found. It is worth noting that phosphorus-solubilizing bacteria can have a positive effect on nitrogen fixation, since the nitrogen fixation process requires a large amount of phosphate ions.

## 4. Discussion

In the literature, selection is often based on multiple criteria, and the best strain is selected based on several parameters. This approach necessitates the management of a large amount of data originating from many strains, since each strain is characterized by several variables. In addition, this approach is resource- and time-consuming. The main challenge is to reduce the amount of primary data obtained while avoiding significant loss of useful strains. Another important task is to determine inclusion/exclusion criteria in order to reduce the number of samples and select the most interesting microorganisms that will be used for further research with plants.

The authors of the paper cited in [21] proposed a new protocol based on two selection stages. The first screening was carried out at a qualitative level. It assessed four indicators, including ammonium production, siderophore production, P-solubilization, and nitrification; the result was represented in binary code and followed by multivariate analysis (PCA). This approach resulted in the identification of homogeneous groups of isolates from related genera with identical indicators across all the tests.

In our study, after initial qualitative screening, several strains belonging to different species of bacteria were selected, which showed exceptional results. These strains were then subjected to a quantitative assessment of their ability to grow in nitrogen-free media; produce siderophores, IAA, and ammonium; and show P-solubilization and antifungicidal activity. Using PCA, bacteria clusters were identified, and one of them was selected due to a high rate of improvement in nutritional availability (e.g., nitrogen, phosphorus) and production of phytohormones, while the second was represented by bacteria exhibiting high levels of production of siderophores and antifungicides. There was also a bacteria cluster with low indicators for both characteristics. This confirms the thesis that it is impossible to find a superorganism that is optimal in terms of all characteristics important for plants. Unfortunately, within the factors that divided the strains into clusters, individual indicators were not distinguished, since all the examined parameters were included in the main components at approximately the same level.

Meanwhile, closely related bacteria could end up in different clusters, which indicates significant diversity of traits useful for plants within the genus. Thus, in this study, the bacteria of various species, such as *Pseudomonas kitaguniensis*, *Pseudomonas koreensis*, *Rahnella aceris*, *Enterobacter ludwigii*, and *Rothia endophytica*, were included in a cluster with high rates of improvement in nutritional availability (nitrogen, phosphorus) and production of phytohormones. The cluster with high levels of antifungicidal activity and siderophore production was represented by bacteria of the species *Enterobacter amnigenus* and *Pseudomonas silesiensis*.

*Enterobacter* spp. is known to have a wide range of PGPB characteristics related to nitrogen fixation, solubilization of phosphorus in soil, production of antibiotics, ability to secrete siderophores, chitinase, ACC deaminase, hydrolytic enzymes besides exopolysaccharides, and increasing soil porosity [22]. *Enterobacter ludwigii* has been reported to show significant IAA production and phosphate solubilization [23], heavy metal resistance [24], and antifungicidal activity [25]. *Enterobacter amnigena* isolates were able to produce ammonia, fix N2, biosynthesize phytohormones, and have a generally beneficial effect on wheat and tomato growth [26], as well as to possess phosphate-solubilizing activity [27].

Many strains of Pseudomonas are beneficial rhizobacteria which promote plant growth. The ability of *Pseudomonas* sp. to enhance plant growth has been attributed to various mechanisms, such as phytohormone production, nutrient solubilization, HCN, siderophore secretion, and biocontrol activity [28]. *Pseudomonas koreensis* is a species whose representa-

tives are repeatedly mentioned in relation to the possibility of nitrogen fixation and phytohormone production [29], as well as phosphate solubilization, ammonium, and siderophore production [30]. *Pseudomonas kitaguniensis* has been described as a phytopathogen causing blight of *Allium fistulosum* L.; however, it is very closely related to other species of the *P. fluorescens* subgroup [31], among which many species of bacteria belong to PGPB.

Rahnella strains are widespread and adapted to a variety of ecological environments that can be related to their resistance to acids, salts, selenium, antibiotics, and heavy metals. *Rahnella aquatilis* is the species most often described in relation to plants, but *Rahnella aceris* is very similar to this species [32]. For *Rahnella aceris*, the properties of plant protection from pathogens [33] and phosphate-solubilizing activity [34] have been described.

Regarding *Rothia endophytica*, the authors could not find articles indicating its positive role in plant development, except for the fact that it was isolated from healthy roots of *Dysophylla stellata* (Lour.) Benth [35]. In the present investigation, this isolate showed high growth ability on a nitrogen-free medium, ammonium production, and phosphate-mobilizing abilities.

Regression analysis conducted to assess the significance of the factors obtained in the cluster analysis revealed a significant effect of factor 1 on the increase in plant height and ear length, as well as on the ammonium content in the soil where bacteria with high indicators of factor 1 (modulo) were introduced. However, it is worth noting that there were also individual examples of the positive effects of strains belonging to other clusters on plant growth. This suggests that additional factors should be considered for strain selection, rather than limiting the selection solely to those that contribute to factor 1.

In the meantime, the cluster analysis showed that it is advisable to search for strains with significant abilities in terms of either providing nutrition and promoting plant growth or resisting phytopathogens. It is rather obvious that a combination of these two types will be successful when forming consortia for applications under natural conditions.

A limitation of this study is related to the possible loss of "interesting" strains, as only 17 strains from various species could be sampled as a consequence of resource constraints. In addition, assessing the significance of the "antimicrobial activity" factor seems insufficient within the framework of the "potted" experiment. It is possible that, in the soil system, their ability to suppress phytopathogens can significantly improve plant growth indicators.

## 5. Conclusions

The following conclusions can be drawn from the results of the conducted experiments:

(1) It was not possible to identify 1–2 of the most important indicators for the primary selection of strains beneficial for plant growth. To effectively search for beneficial bacterial growth strains, it is insufficient to use only indicators such as growth on nitrogen-free media, phosphate mobilization, ammonium production, and auxin production. The development of additional effective and simple methods for screening other bacterial properties that may be beneficial to plants (e.g., ACC deaminase, enzymes, etc.) is required.

(2) The selection of strains should be carried out independently based on the characteristics of "plant nutrition" and "antimicrobial activity". Furthermore, bacteria that exhibit the maximum values of these factors can be combined into consortia.

(3) Characteristics beneficial for plant growth can be revealed in almost any species of soil microorganism. In such a way, a strain of *Rothia endophytica* was first discovered which showed high growth ability on a nitrogen-free medium, ammonium production, and phosphate-mobilizing abilities.

**Supplementary Materials:** The following supporting information can be downloaded at: https://www.mdpi.com/article/10.3390/microbiolres15010002/s1, Table S1: Description of soil samples, morphology of colonies and bacteria, phylogenetic trees for 17 strains participating in the experiment on the effect on plant growth.

**Author Contributions:** Conceptualization, E.V. and N.S.; evaluation of the properties of bacteria useful for plants, E.V. and I.T.; statistical data analysis, E.S.; work with microorganisms, I.T.; sequencing, O.M.; construction of phylogenetic trees, descriptions of microorganisms, I.H.; evaluation of antifungicidal activity, M.M.; work with plants, O.S.; soil analysis, M.B. and I.I.; organization of sample collection through the "civil science", M.G. All authors have read and agreed to the published version of the manuscript.

**Funding:** This work was supported by the Ministry of Science and Higher Education of the Russian Federation, agreement No. 075-15-2021-1085.

**Institutional Review Board Statement:** Not applicable.

**Informed Consent Statement:** Not applicable.

**Conflicts of Interest:** The authors declare that the research was conducted in the absence of any commercial or financial relationships that could be construed as a potential conflict of interest.

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
