# Peer review of "Properties of Potential Plant-Growth-Promoting Bacteria and Their Effect on Wheat Growth Promotion (Triticum aestivum) and Soil Characteristics"

_2036-7481, doi:10.3390/microbiolres15010002_

Round 1
Reviewer 1 Report
Comments and Suggestions for Authors
The authors need to determine the core questions to be solved in this study, such as: finding germplasm resources beneficial to plant growth, or solving the technical problems of screening beneficial bacteria for plant growth? On this basis:
1. Rewrite the introduction: please focus on the core scientific questions to be solved and pay attention to the logical of the language. "Citizen science" needs to be explained.
2. Materials and methods:
1) No matter the soil samples collected by the team, or the strains obtained directly through the "citizen science" program, the specific information of the source of the strains and the soil samples should be provided.
2) How are the bacteria obtained from the "Citizen science" project preserved? Is there a need for resurrection?
3) When detecting each index for each strain of bacteria, is the inoculation amount unified? If it's not uniform, the results can't be compared.
4) Why only one pathogenic fungus was selected in the test of antifungal activity? Was Fusarium oxisporum the standard strain?
5) In a pot experiment, is the selected soil sterilized? If not treated, will the experimental results be affected by other uncertain factors?
3. Results:
1) The description of the results of strain identification is rather general, and it is recommended to add the data of all colonies, Gram staining and other measurement indicators in the form of attachments.
2) To add phylogenetic trees at least in the text of the results.
3) All data appearing in the text list should be three repeated average ± standard deviation.
4) Should be standardized, unified text of the graph, table form, such as: Figure 3.
5) Some of the measurement indicators in the materials and methods part are not presented in the results part.
4. Discussion: after determining the core scientific questions, the discussion should be based on the data of this study. It can't be generalized; it should be focused on the core scientific questions.
5. Conclusion: need to be based on the core scientific questions, research data and discussion.
6. References: need to be carefully checked, such as [20], [10].
Reviewer 2 Report
Comments and Suggestions for Authors
The present study incorporates bacterial isolation and identification, pot plant growth assessment, and soil characteristic evaluation. The research yielded a substantial amount of data, which can be valuable for researchers in this field. However, there are a few comments that need to be addressed:
1. The introduction part should provide more clarity on the specific objective of this study, considering the existence of numerous similar works already published.
2. Carefully review the text throughout the manuscript to ensure proper formatting of subscripts (such as chemical formulas) and superscripts (in scientific counting methods) according to the paper format.
3. In the pot trial section, it is unclear why the bacteria were not inoculated directly into the soil. The involvement of bacteria seems to be limited to pretreating the seeds, making it challenging to assess their direct impact on plant growth and soil characteristics.
4. It is recommended to use seedlings instead of seeds for a more consistent sample, as seeds can produce different types of seedlings.
5. Clarify the criteria used for selecting the 17 strains of bacteria in the study.
6. Include the standard error values for the analyses presented in Table 1.
7. It is advisable to improve the resolution of Figure 3 for better visibility. Additionally, consider performing statistical analyses for all the data presented.
8. Consider conducting correlation analyses between the characteristics of the 17 bacterial strains, plant growth, and soil indicators to gain further insights.
Reviewer 3 Report
Comments and Suggestions for Authors
This paper reports the growth promoting bacteria from Russian soil collected by many students in school. This paper has value to be published. But there are many mistakes in English typing, Figures and Table should be revised.
I add comments directly on the PDF file. Please check and revise.

Author Response
Thank you for carefully reading the article and your comments. We tried to make corrections according to your recommendations.
Only one remark caused a difficulty. Subcultivation is a standard method, to which it is difficult to choose the first authorship.
Round 2
Reviewer 1 Report
Comments and Suggestions for Authors
The quality of the paper has been improved to some extent after revision, but there are problems with the treatment of soil in the part of the method: because a single soil sterilization is not enough to remove spore containing bacteria, at least two or three sterilization treatments (and also sterilization intervals to allow spores to germinate) to completely remove spore-bearing bacteria.The unremoved bacteria in the soil can affect the evaluation of strain traits.
1. Figure 3 Repeated figures b and c;
2. The amendments to the discussion section are very limited and there are no real changes.
Author Response
1. Clarified the parameters of soil sterilization
2. These graphs are similar, but not the same
Reviewer 2 Report
Comments and Suggestions for Authors
1. Throughout the manuscript, please use consistently ml or mL, μl or μL in units but do not mix the two. Also, h or hrs for time.
2. “10 mkl of bacterial solution”, you mean 10 mL?
3. “During the growing period, the maximum temperature was 26, the minimum was 3.7, and the average was 17.1.” Add the unit for the temperature.
4. The strains number in all tables and figures needs to be consistent.
5. In Figure 3, the difference analysis between samples also needs to be clarified. The figure caption should be below the figure, not above it. Why no static analysis in Fig.b and Fig.d?
Author Response
- We have made corrections
- Removed this sentence. Inoculation was carried out in the solution of bacteria specified in the sentence above.
- We have made corrections
- We have made corrections
- The analysis of differences was carried out using the Student's criterion, this is indicated in the "Materials and Methods". No statistically significant differences from the control were found for graph b. No statistical calculation was carried out for the graph d. , since a single measurement was carried out (due to the low mass of dry grain). This is indicated in the text.
Reviewer 3 Report
Comments and Suggestions for Authors
Most of the comments were well revised. I agree this paper will be accepted in this Journal.
Author Response
Thanks for the valuable comments that helped improve the article